# LLEOT: A Privacy-Enhancing Offsite Tuning Framework via Loss Landscape Elevation

## Abstract

Adapting large language models (LLMs) to domain-specific tasks via fine-tuning is often infeasible: model parameters are protected by intellectual property, while sensitive data cannot be shared due to privacy regulations. Offsite Tuning addresses this by training adapters on emulators of the original model, but current emulators retain substantial inference ability, exposing model capability privacy and risking misuse. We propose Loss Landscape Elevation Offsite Tuning (LLEOT), a framework that secures both data and model capability privacy. Its core component, Loss Landscape Elevation (LLE), enforces a fixed loss margin between emulator and model, which we theoretically show (Theorem 1) simultaneously (i) degrades emulator inference through perplexity amplification and (ii) preserves gradient alignment, ensuring consistent convergence of prompt optimization. Combined with Collaborative Prompt Knowledge Distillation (CPKD), our method enables adapters trained on emulators to transfer effectively to the original model. Extensive experiments on the OpenBookQA, SocialIQA, ARC-Challenge, and WebQuestions datasets confirm LLEOT achieves strong adaptation while mitigating emulator misuse.

## 1 Introduction

In the field of natural language processing, fine-tuning pre-trained large language models (LLMs) (Wei et al., 2022; Muennighoff et al., 2023; Liu et al., 2022) on domain-specific data has become a widely adopted technique for adapting general-purpose models to specialized tasks. However, this approach faces significant practical constraints, particularly concerning intellectual property and data privacy (Gupta et al., 2022; Lyu et al., 2024). On the one hand, due to proprietary protections and licensing restrictions, many high-performing LLMs cannot be openly distributed to external data owners for fine-tuning (Li et al., 2023). On the other hand, even when model owners offer data submission interfaces for cloud-based training, stringent privacy regulations in fields such as healthcare (Nguyen et al., 2022) and finance (Kang et al., 2024; Oualid et al., 2025) often prohibit the upload of sensitive data to third-party services. This fundamental conflict, where neither the model nor the data can be shared, creates a significant barrier to effective model adaptation, leaving valuable private data untapped and limiting the applicability of closed-source models in critical domains.

A promising approach is to construct a privacy-preserving *emulator* of the original model to serve as a bridge for knowledge transfer. As shown in Figure 1(a), data owners use this *emulator* to locally train an *adapter* that encodes the knowledge from their domain-specific data. This adapter is then returned to the model owner to be applied to the original model, enabling the model to acquire knowledge from the data without exposing the model parameters or the data itself. Xiao et al. (2023) first introduced this method, naming it Offsite Tuning, which constructs an emulator through model compression and knowledge distillation. FedBiOT (Wu et al., 2024) extended this approach to a federated setting and replaced the adapter with LoRA (Hu et al., 2022). CRaSh (Zhang et al., 2023a) accelerates emulator construction by substituting knowledge distillation with layer importance-based selection, where high-importance layers replace low-importance ones. These methods employ techniques such as knowledge distillation (Mora et al., 2024; Huang et al., 2024) to align the emulator with the original model, ensuring that the adapter trained on the emulator remains applicable to the original model. However, this approach results in an emulator that retains a significant portion of the original model's inference capabilities (Figure 1(c)), which inadequately protects the model's

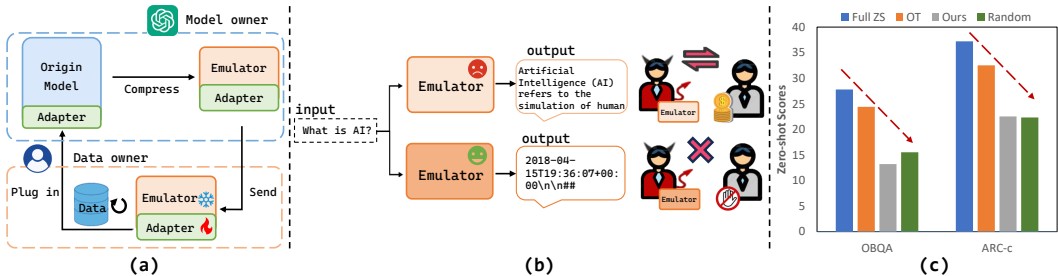

Figure 1: (a) Overview of privacy-preserving emulator methods. (b) Illustration of model capability leakage. (c) Comparison of inference ability on OBQA (Mihaylov et al., 2018) and ARC-c (Clark et al., 2018) dataset: Full ZS (original LLM Qwen2-1.5B-Instruct (Yang et al., 2024)) and OT (80% compressed LLM) achieve high scores, indicating privacy leakage, while our method and Random (randomly initialize model) show lowscores, demonstrating effective protection of model privacy.

capability privacy. Consequently, malicious data owners could potentially use this emulator to extract the model's knowledge or engage in unauthorized activities, thereby infringing upon the model owner's intellectual property rights, as shown in Figure 1(b).

To address the above challenge, we propose Loss Landscape Elevation Offsite Tuning (LLEOT), a novel framework that extends privacy protection to model capability privacy. The core of LLEOT lies in Loss Landscape Elevation (LLE). Specifically, we adjust the emulator to have a consistently higher loss than the original model by a fixed margin across all data points (Section 4.2). Though simple, our approach offers two key advantages as proven in Theorem 1. First, the elevated loss disables the emulator's inference ability, preserving the original model's capability privacy. Second, it maintains geometric consistency between the loss landscapes (see Figure 3), keeping the adapter's loss gradients coherent across models. This ensures adapters optimized on the emulator perform well when transferred to the original model. In theory, our method is applicable to various types of adapters. In this paper, we focus on soft prompts for their computational efficiency and ease of optimization. Additionally, to further enhance gradient consistency between the emulator and the original model, we first align the emulator and original model using our proposed Collaborative Prompt Knowledge Distillation (CPKD), a knowledge distillation technique tailored for soft prompts (see Section 4.1), before performing LLE. Our contributions can be summarized as follows:

- We identify the overlooked risk of model capability privacy in Offsite Tuning: existing emulators retain substantial inference power, enabling malicious data owners to extract proprietary knowledge or misuse the model.

- We propose Loss Landscape Elevation Offsite Tuning (LLEOT), which applies Loss Landscape Elevation (LLE) to disable emulator inference while preserving gradient alignment with the original model. We provide a theoretical guarantee (Theorem 1) that LLE both amplifies emulator perplexity and preserves convergence to the same optimal prompt.

- We integrate LLE with Collaborative Prompt Knowledge Distillation (CPKD)—a distillation strategy tailored for soft prompts—and show that adapters optimized on the emulator transfer effectively to the original model.

- Comprehensive experiments demonstrate the superiority of our proposed LLEOT. It achieves better privacy protection while maintaining higher model performance than existing methods.

## 2 RELATED WORKS

**Large Language Models.** Through pre-training on massive corpora, large language models (LLMs) (Kojima et al., 2022a; Kung et al., 2023; Wang et al., 2024a) have acquired extensive general knowledge and demonstrated remarkable performance across a wide range of natural language processing tasks, often effectively addressing these tasks via zero-shot (Kojima et al., 2022b; Ji et al., 2024) learning. However, when applied to domain-specific problems, LLMs still require

fine-tuning (Bai et al., 2024; Zhang et al., 2024) on relevant data to better adapt to the target tasks. Unfortunately, in many real-world scenarios, the model and the data are owned by different parties, and fine-tuning through mutual sharing is often infeasible for reasons including intellectual property protection. Black-box tuning (Yu et al., 2023; Zheng et al., 2024) approaches upload data to the model owner and adjust parameters based on output text, which helps protect the privacy of the LLMs but poses risks to user data. Alternative methods, such as federated learning (McMahan et al., 2017; Shi & Radu, 2021) and split learning (Li et al., 2024), distribute the model to data owners to avoid data transmission, yet these approaches expose model privacy. Given the high value of large language models, such solutions are often unacceptable to model owners.

**Privacy-preserving fine-tuning of large language models.** To jointly protect model privacy and data privacy during fine-tuning, Offsite Tuning (OT) (Xiao et al., 2023) compresses the original model and applies knowledge distillation to obtain an emulator and an adapter; the data owner fine-tunes the adapter with the help of the emulator and then returns the tuned adapter to the model owner for integration into the original model. Building on this idea, Fedbiot (Wu et al., 2024) extends OT to a federated setting and employs LoRA adapters to further reduce communication overhead. In contrast, CRaSh (Zhang et al., 2023a) constructs an emulator without knowledge distillation by performing layer importance ranking on the original model and replacing less important layers with repeated high-importance layers. While these approaches safeguard model parameter privacy and data privacy, their reliance on knowledge distillation or importance-based layer selection causes the emulator to inherit part of the original model's reasoning capability. Consequently, data owners may use the emulator to produce inference results similar to those of the original model, leading to capability privacy leakage and incomplete protection of model privacy.

## 3 PROBLEM FORMULATION

**Privacy Requirements.** We consider a scenario involving two parties: a model owner and a data owner. The model owner possesses a closed-source LLM and provides only paid query-based access to the data owner, without sharing the original model or any substitute model that exhibits comparable inference capabilities. The data owner holds a private dataset and aims to tune the LLM on this data to address their downstream task, while ensuring that their data privacy remains protected from the model owner.

**Setup.** Given a original model $\mathcal{M}$ parameterized by $\Theta$ and a downstream dataset $D$, let $\mathcal{M}_{\Theta+\Delta}$ denote the result of directly fine-tuning $\mathcal{M}_{\Theta}$ on $D$ with an adapter, where $\Delta$ represents the adapter parameters. To simultaneously protect model privacy and data privacy, we aim to identify an emulator $\mathcal{E}$ parameterized by $\Theta^*$ that is smaller and weaker than $\mathcal{M}_{\Theta}$, which serves as a proxy for the original model to be provided to the data owner for fine-tuning. The architecture and parameters of this emulator should differ from those of the original model, and its inference capability should approximate that of a randomly initialized model $\mathcal{M}_{\mathcal{R}}$. This ensures that the data owner can neither access the parameters of the original model nor leverage the emulator for activities that infringe upon the model owner's intellectual property, such as repackaging it as their own closed-source model. The data owner then performs adapter fine-tuning on the emulator using $D$ to obtain $\mathcal{E}_{\Theta^*+\Delta^*}$, where $\Delta^*$ denotes the adapter parameters learned on the emulator. We require that transferring the fine-tuned adapter weights $\Delta^*$ back to the original model (i.e., forming $\mathcal{M}_{\Theta+\Delta^*}$) should yield performance comparable to that achieved by directly fine-tuning $\mathcal{M}$ (i.e., $\mathcal{M}_{\Theta+\Delta}$), without requiring access to $\mathcal{M}$ itself.

**Metrics**. We introduce Capability Privacy Leakage (CPL), a new metric to quantify the capability leakage from the original model through the emulator. It is defined as the ratio of their zero-shot performance scores on a given task:

$$\text{CPL} = \frac{S_{zs}(\mathcal{E})}{S_{zs}(\mathcal{M})} \times 100\%, \tag{1}$$

where $S_{zs}(\cdot)$ is the zero-shot score function, $\mathcal{M}$ and $\mathcal{E}$ denote the original model and the emulator, respectively. Capability privacy protection is considered to be in effect only when the CPL value is below $100\%$. A lower CPL value signifies a lesser degree of leakage and thus more effective protection.

Figure 2: Overview of the emulator construction process in the LLEOT Framework. This process involves three key steps. (a) First, we initialize the emulator by applying layerdrop to the original model. (b) Then, we align the emulator with the original model using CPKD to enhance soft prompt transferability. (c) Finally, the core mechanism, LLE, disrupts the emulator's inference capability while preserving the gradient alignment between the two models.

## 4 METHODOLOGY

As shown in Algorithm 1, the workflow of our proposed LLEOT framework comprises three phases: (1) Emulator Construction: The model owner constructs an emulator and sends it to the data owner; (2) Adapter Training: The data owner fine-tunes the adapter on the emulator using local data; and (3) Adapter Transfer: The model owner incorporates the fine-tuned adapter, returned by the data owner, into the original model. In this work, we specifically focus on the implementation where adapters are soft prompts, due to their computational efficiency. The core of our method lies in the emulator's construction process, which is illustrated in Figure 2. The process involves three key steps: first, we initialize the emulator by randomly discarding a certain proportion of layers from the original model. Then, we align the emulator with the original model using our proposed Collaborative Prompt Knowledge Distillation (CPKD). Finally, we disrupt the emulator's inference capability through the proposed Loss Landscape Elevation (LLE) technique while preserving the alignment between the two models.

### 4.1 COOPERATIVE PROMPT KNOWLEDGE DISTILLATION

The emulator, initialized by discarding layers from the original model, inevitably exhibits discrepancies that make the adapter trained on it difficult to apply to the original model. To address this issue, methods such as OT (Xiao et al., 2023) align the two models through knowledge distillation (Hinton et al., 2015), with the loss function expressed as:

$$\mathcal{L}_{KD} = \mathbb{E}_{x \sim \mathcal{X}_d} ||(\mathbf{H}_{\mathcal{E}}^{(-1)}(x), \mathbf{H}_{\mathcal{M}}^{(-1)}(x))||^2, \tag{2}$$

where $\mathcal{X}_d$ is the distillation dataset, and the notation $\mathbf{H}^{(-1)}$ represents the hidden state extracted from the final transformer layer. The subscripts $\mathcal{E}$ and $\mathcal{M}$ refer to the emulator and the original model, respectively. The term in the parentheses, e.g., $(x)$, indicates the input provided to the model.

This approach, however, fails when using soft prompts as adapters. Unlike discrete tokens, soft prompts are vectors optimized in a continuous representation space. Traditional knowledge distillation aligns models only at discrete token instances, neglecting the broader continuous space. As a result, the emulator's learned soft prompt may occupy a misaligned position within this space, rendering its transfer to the original model problematic.

To address this challenge, we propose the Proxy Prompt Distillation Loss to align the continuous representation spaces of the emulator and the original model. We use randomly initialized soft prompts as proxies for the real soft prompts, prepending them to the distillation data. We then align the portions of the feature representations corresponding to the distillation data, which are generated by the emulator and the original model from the concatenated input. This loss can be formulated as:

$$\mathcal{L}_{PPD} = \mathbb{E}_{x \sim \mathcal{X}_d, P' \sim \mathcal{N}(\mu, \sigma^2)} ||(\mathbf{H}_{\mathcal{E}, L_p:}^{(-1)}(P', x), \mathbf{H}_{\mathcal{M}, L_p:}^{(-1)}(P', x))||^2, \tag{3}$$

where $P'$ is the proxy soft prompt, $L_p$ denotes its length, and $\mathcal{N}(\mu, \sigma^2)$ is a normal distribution with mean $\mu$ (e.g., 0) and standard deviation $\sigma$ (e,g., 20), determined experimentally.

---

**Algorithm 1** LLEOT

---

**Input:** Original model $\mathcal{M}$, distillation dataset $\mathcal{X}_d$, elevation dataset $\mathcal{X}_e$, private local datasets $\mathcal{D}_p$, hyperparameters $w_1, w_2, w_3$, elevation margin $H$, dropout rate $\beta$

1: **Model owner**
2: **Stage 1: LayerDrop**
3: $\mathcal{E} \leftarrow$ LayerDrop($\mathcal{M}, \beta$)
4: **Stage 2: Cooperative Prompt Knowledge Distillation (CPKD)**
5: **for** each batch $x \sim \mathcal{X}_d$ **do**
6:     Randomly sample proxy soft prompt $P' \sim \mathcal{N}(\mu, \sigma^2)$
7:     Optimize $\mathcal{E}$ with respect to Equation (5)
8: **end for**
9: **Stage 3: Loss Landscape Elevation (LLE)**
10: **for** each batch $x \sim \mathcal{X}_e$ **do**
11:     Randomly sample proxy soft prompt $P' \sim \mathcal{N}(\mu, \sigma^2)$
12:     Optimize $\mathcal{E}$ with respect to Equation (7)
13: **end for**
14: $\mathcal{E}^* = \mathcal{E}$
15: Model owner sends $\mathcal{E}^*$ to Data owner
16: **Data owner**
17: **Prompt Tuning for Downstream Tasks**
18: Initialize soft prompt $P$
19: **for** each batch $(x, y) \sim \mathcal{D}_p$ **do**
20:     Compute downstream task loss: $\mathcal{L}_{ds}$
21:     Update prompt: $P \leftarrow P - \eta \nabla_P \mathcal{L}_{ds}$
22: **end for**
23: $P^* = P$
24: Data owner sends $P^*$ to Model owner
25: **return** Original model with optimized soft prompt $\{\mathcal{M}, P^*\}$

---

Additionally, following OT, we incorporate a language modeling loss when optimizing the emulator. Let $n$ be the number of tokens in an input text $x$. The loss $\mathcal{L}_{LM}$ can be expressed as:

$$\mathcal{L}_{LM} = -\frac{1}{n}\sum_{i=1}^{n} \log p_{\mathcal{E}}(x_i | x_{1:i-1}). \tag{4}$$

Here, $p_{\mathcal{E}}(x_i | x_{1:i-1})$ denotes the probability of the emulator correctly predicting the $i$-th token given the preceding $i - 1$ tokens.

Finally, the overall objective of CPKD for distilling the emulator can be expressed as:

$$\mathcal{E}^* = \arg\min_{\mathcal{E}} w_1 \mathcal{L}_{LM} + w_2 \mathcal{L}_{PPD} + w_3 \mathcal{L}_{KD}, \tag{5}$$

where $w_1$, $w_2$ and $w_3$ are hyperparameters used to balance the contributions of each term.

### 4.2 Loss Landscape Elevation

As discussed in Section 4.1, we apply CPKD to the emulator to enhance the transferability of fine-tuned soft prompts to the original model. However, a side effect of this process is that the emulator inevitably inherits part of the original model's knowledge and reasoning capabilities. This creates a potential privacy risk: providing the distilled emulator to the data owner may enable them to extract proprietary knowledge (Chua et al., 2024; Dong, C. and Xie, Y. and Ding, B. and others, 2023; Wang et al., 2024b) or even repackage the emulator as a commercial product (Jagarlamudi et al., 2024).

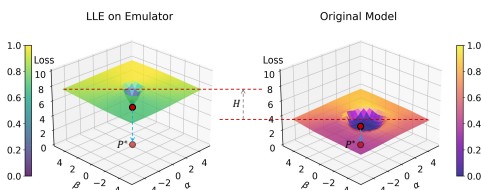

Figure 3: Visualization of LLE shows the loss landscapes of the emulator (left) and the original model (right) in the same soft prompt parameter space ($\alpha\beta$-plane).

To mitigate this risk, we propose *Loss Landscape Elevation* (LLE), a method designed to impair the emulator's reasoning capabilities while preserving its gradient guidance for soft prompt tuning. The core idea is to uniformly elevate the emulator's loss landscape while aligning its geometry with that of the original model. Specifically, for any soft prompt $P$, input text $x$, we enforce

$$\mathcal{L}_{\mathcal{E}}(P; x) = \mathcal{L}_{\mathcal{M}}(P; x) + H, \tag{6}$$

where $\mathcal{L}_{\mathcal{E}}$ and $\mathcal{L}_{\mathcal{M}}$ denote the prompt tuning loss for the emulator and the original model, respectively. $H \geq 0$ is a hyperparameter for the fixed loss margin. More formally, the objective of LLE is formulated as:

$$\mathcal{E}^* = \arg\min_{\mathcal{E}} \mathbb{E}_{x \sim \mathcal{X}_e, P' \sim \mathcal{N}(\mu, \sigma^2)} |\mathcal{L}_{\mathcal{E}}(P'; x) - \mathcal{L}_{\mathcal{M}}(P'; x) - H|, \tag{7}$$

where $\mathcal{X}_e$ denotes the elevation dataset.

Below we prove that LLE can preserve model privacy. We expand the prompt tuning loss into the following expression:

$$\mathcal{L}_{\mathcal{E}}(P; x) = -\frac{1}{n}\sum_{i=1}^{n} \log(p_{\mathcal{E}}(x_i|P, x_{1:i-1})) = -\frac{1}{n}log(\hat{p}_{\mathcal{E}}(x|P)), \quad \hat{p}_{\mathcal{E}}(x|P) = \prod_{i=1}^{n} p_{\mathcal{E}}(x_i|P, x_{1:i-1}), \tag{8}$$

Here, $p_{\mathcal{E}}(x_i|P, x_{1:i-1})$ denotes the probability of the emulator correctly predicting the $i-th$ token given the soft prompt and the preceding $i-1$ tokens, and $\hat{p}_{\mathcal{E}}(x|P)$ represents the joint probability of the emulator predicting the entire sequence $x$ given the prompt $P$.

As the LLM perplexity is defined as $\hat{p}^{-1/n}$, and given that our LLE method enforces a fixed margin between the losses of the original model and the emulator, as described in Equation 6, we can derive the following relationship between their perplexities:

$$\mathrm{PPL}_{\mathcal{E}} = e^H \cdot \mathrm{PPL}_{\mathcal{M}}. \tag{9}$$

Clearly, the perplexity of the emulator $\mathrm{PPL}_{\mathcal{E}}$ is significantly higher than that of the original model $\mathrm{PPL}_{\mathcal{M}}$, which demonstrates the model privacy protection capability of LLE.

**Theorem 1** (Effect of LLE on Emulator). *For the emulator $\mathcal{E}$ constructed with Loss Landscape Elevation (LLE), we have*

$$\mathrm{PPL}_{\mathcal{E}} = e^H \cdot \mathrm{PPL}_{\mathcal{M}} \quad \text{and} \quad \nabla_P \mathcal{L}_{\mathcal{E}}(P; x) = \nabla_P \mathcal{L}_{\mathcal{M}}(P; x), \tag{10}$$

*where $\mathrm{PPL}_{\mathcal{E}}$ and $\mathrm{PPL}_{\mathcal{M}}$ denote the perplexities of the emulator and the original model on input $x$, respectively.*

This theorem proved in Appendix D demonstrates that LLE exponentially increases the emulator's perplexity, thereby degrading its inference capability, while leaving the gradient landscape unchanged. Consequently, gradient-based optimization converges to the same optimal soft prompt $P^\star$ as in the original model, ensuring effective prompt transfer despite impaired emulator reasoning.

### 4.3 Prompt Tuning

Upon completion of the emulator, the model owner sends it to the data owner. The data owner optimize a soft prompt $P$ on their private dataset $\mathcal{D}_p$ by minimizing the downstream task loss $\mathcal{L}_{ds}$:

$$P^* = \arg\min_{P} \mathbb{E}_{(x,y) \sim \mathcal{D}_p}[\mathcal{L}_{ds}(\mathcal{E}; P, x, y)] \tag{11}$$

The resulting prompt, $P^*$, is then sent back to the model owner, where the prompt is integrated into the original model to adapt it for the downstream task.

Furthermore, our findings in Appendix B.2 show that LLEOT is orthogonal to data privacy strategies. This means the fine-tuned prompt can be sanitized before being sent back, safeguarding the local data's privacy against various inference attacks from the model owner, such as membership inference (Duan et al., 2023), all without significantly compromising the prompt's utility.

Table 1: Comparative experiment results. 'Acc' denotes accuracy (higher is better), and 'CPL' represents the model capability privacy measure (lower values indicate better protection). DR stands for dropout rate. For each DR setting, the best results are in **bold**, and the second best are underlined.

| DR | Method | OBQA Acc(↑) | OBQA CPL(↓) | SIQA Acc(↑) | SIQA CPL(↓) | ARC-c Acc(↑) | ARC-c CPL(↓) | WebQs Acc(↑) | WebQs CPL(↓) |
|---|---|---|---|---|---|---|---|---|---|
| | | | | | Qwen2-1.5b | | | | |
| | Full ZS | 27.80 | 100.00 | 46.47 | 100.00 | 37.20 | 100.00 | 1.82 | 100.00 |
| - | Full PT | 35.80 | 100.00 | 54.52 | 100.00 | 41.98 | 100.00 | 30.73 | 100.00 |
| | Random | 14.13 | 55.83 | 35.57 | 71.72 | 22.10 | 60.00 | 0.33 | 0.00 |
| | OT | 33.80 | 89.45 | 51.03 | 96.76 | 38.54 | 87.62 | 26.62 | 220.95 |
| 0.2 | CRaSh | 31.20 | 59.71 | 50.10 | 79.62 | 39.50 | 76.83 | 27.17 | 59.34 |
| | Ours | **33.87** | **45.56** | **53.34** | **75.19** | **41.98** | **60.08** | **28.76** | **0.00** |
| | OT | 27.20 | 70.02 | 46.80 | 86.90 | 37.29 | 66.34 | 21.65 | 166.59 |
| 0.5 | CRaSh | 24.67 | 54.68 | 48.00 | 76.35 | 39.33 | 58.71 | 18.16 | 0.00 |
| | Ours | **34.20** | **46.52** | **50.04** | **75.87** | **40.44** | **48.39** | **24.15** | **0.00** |
| | | | | | Gemma2-2b | | | | |
| | Full ZS | 35.60 | 100.00 | 50.00 | 100.00 | 50.85 | 100.00 | 8.07 | 100.00 |
| - | Full PT | 45.80 | 100.00 | 56.60 | 100.00 | 54.30 | 100.00 | 38.09 | 100.00 |
| | Random | 33.58 | 47.87 | 46.69 | 66.34 | 45.30 | 43.07 | 0.33 | 0.00 |
| | OT | 41.73 | 85.39 | 56.29 | 87.76 | 44.97 | 68.40 | 26.62 | 48.17 |
| 0.2 | CRaSh | 41.20 | 51.69 | 55.22 | 77.58 | 48.72 | 47.83 | **34.25** | 5.45 |
| | Ours | **45.33** | **37.45** | **56.94** | **69.13** | **54.47** | **39.72** | 28.17 | **0.00** |
| | OT | 39.00 | 74.91 | 50.05 | 82.57 | 38.40 | 56.61 | **25.26** | 27.49 |
| 0.5 | CRaSh | 35.80 | 48.31 | 51.50 | 70.66 | 49.57 | 43.46 | 19.47 | 3.08 |
| | Ours | **44.87** | **38.01** | **55.01** | **70.15** | **52.56** | **39.76** | 22.44 | **0.00** |
| | | | | | Llama3.2-3b | | | | |
| | Full ZS | 28.20 | 100.00 | 45.04 | 100.00 | 43.69 | 100.00 | 11.32 | 100.00 |
| - | Full PT | 36.11 | 100.00 | 56.42 | 100.00 | 48.12 | 100.00 | 36.88 | 100.00 |
| | Random | 12.91 | 52.34 | 33.96 | 74.05 | 19.45 | 51.48 | 0.00 | 0.00 |
| | OT | 32.40 | 87.00 | 45.67 | 94.88 | 43.20 | 80.32 | 25.13 | 78.55 |
| 0.2 | CRaSh | 31.00 | 67.38 | **49.38** | 88.06 | 43.23 | 75.19 | 23.92 | 2.65 |
| | Ours | **33.60** | **41.37** | 49.02 | **76.90** | **45.62** | **45.16** | **26.13** | **0.00** |
| | OT | 29.47 | 77.30 | 43.30 | 91.05 | 39.62 | 60.54 | **23.90** | 36.26 |
| 0.5 | CRaSh | 26.07 | 65.25 | 47.50 | **77.38** | 43.69 | 52.92 | 21.67 | **0.00** |
| | Ours | **33.40** | **53.43** | **48.21** | 81.25 | **46.96** | **47.19** | 15.45 | **0.00** |

# 5 EXPERIMENTS

## 5.1 EXPERIMENTAL SETUP

**Models and Datasets.** We evaluate our method on three LLMs: Qwen2-1.5B-Instruct (Yang et al., 2024), Gemma-2-2b-it (Team et al., 2024), and Llama-3.2-3B-Instruct (Touvron et al., 2023). We consider two dropout rates, 0.2 and 0.5, which represent the ratio of layers dropped from the original model when initializing the emulator. Experiments are conducted on four question-answering benchmark datasets: OpenBookQA (Mihaylov et al., 2018), SocialIQA (Sap et al., 2019), ARC-Challenge (Clark et al., 2018), and WebQuestions (Berant et al., 2013). More experimental details are provided in Section A.4.

**Baseline Methods.** We compare our approach with the following five methods: (1) **Full ZS**: Zero-shot performance of the original model, representing the lower bound that our method should improve upon. (2) **Full PT**: Prompt tuning directly on the original model using the downstream dataset. While serving as a theoretical upper bound for transfer performance, this is impractical in real scenarios due to privacy concerns. (3) **Random**: A model with the same architecture as the original

Table 2: Results of ablation experiments for the LLEOT emulator construction phase on Qwen2-1.5B-Instruct, with dropout rates of 0.2 and 0.5. CPKD refers to Collaborative Prompt Knowledge Distillation phase, and LLE to Loss Landscape Elevation phase. The symbols ✓ and ✗ respectively indicate the inclusion and ablation of the corresponding settings. Best in **bold**.

|   | CPKD | LLE | DR=0.2 | | DR=0.5 | |
|---|---|---|---|---|---|---|
|   |   |   | Acc($\uparrow$) | CPL($\downarrow$) | Acc($\uparrow$) | CPL($\downarrow$) |
| 1 | ✓ | ✓ | **33.87** | 45.56 | 34.20 | **46.52** |
| 2 | ✗ | ✓ | 33.60 | **43.88** | 23.00 | 48.92 |
| 3 | ✓ | ✗ | 33.00 | 87.77 | **35.40** | 74.10 |
| 4 | ✗ | ✗ | 31.20 | 74.10 | 24.40 | 58.27 |

model but with randomly initialized weights. This represents a theoretical upper bound for model capability privacy protection. (4) **Offsite Tuning (OT)** (Xiao et al., 2023): The first work to propose the offsite fine-tuning approach based on emulator construction. It utilizes LayerDrop and knowledge distillation to build an emulator, which guides the data owner in fine-tuning the adapter without transmitting the original model or private data, demonstrating promising performance. (5) **CRaSh** (Zhang et al., 2023b): An OT variant that constructs the emulator via layer-importance selection instead of knowledge distillation. It was the prior state-of-the-art method among open-source OT approaches.

**Metrics.** We evaluate our method based on two aspects: (1) the performance of the original model after incorporating the emulator-trained weight $\Delta^*$. Since all benchmarks are multiple-choice datasets, we report accuracy for this aspect (for Full ZS, Full PT and Random, we report the model's accuracy directly); and (2) the capability privacy protection of the emulator, which we assess using the Capability Privacy Leakage (CPL) metric, as defined in Section 3. We use `lm-eval-harness` [1] to evaluate our models for a fair comparison.

## 5.2 MAIN RESULTS

To validate the transfer performance and model privacy preservation capabilities of LLEOT, we conducted comparative analyses with baseline methods. The results, averaged over three experimental runs, are presented in Table 1. From the table, we can derive the following insights: 1) LLEOT demonstrates superior performance over existing methods in almost all experimental settings across the three models, in terms of both average accuracy and the CPL measure. Notably, under certain experimental settings, our method achieves a CPL score even lower than that of a randomly initialized model. This strongly suggests that the emulator constructed by our method offers robust model capability privacy protection, while the soft prompts fine-tuned on it remain highly transferable to the original model. 2) The knowledge distillation-based method OT shows some improvement in average accuracy over the importance pruning-based method CRaSh, but it falls short in terms of model capability privacy protection. This suggests that, compared to importance pruning, knowledge distillation causes the emulator to inherit more reasoning capabilities from the original model, leading to more severe capability privacy leakage. 3) The performance of the importance pruning-based method CRaSh surpasses that of OT. However, compared to our LLEOT, it exhibits lower average accuracy and worse CPL scores under most experimental settings. This disparity is attributed to the fact that importance pruning fails to completely impair the emulator's inference capabilities, which limits its effectiveness in capability privacy protection. 4) As the compression ratio decreases, the average accuracy of OT and CRaSh improves, but their model capability privacy protection metric deteriorates. In contrast, LLEOT also exhibits improved average accuracy with reduced compression ratios, while its model capability privacy protection metric remains stable, indicating no additional leakage of model capability privacy.

Table 3: Results of ablation experiments for the LLEOT knowledge distillation strategy on Qwen2-1.5B-Instruct. The dropout rate is set to 0.5. $\mathcal{L}_{PPD}$ and $\mathcal{L}_{KD}$ respectively signifies the traditional distillation loss function and the proxy prompt distillation loss function. Best in **bold**.

| | $\mathcal{L}_{LM}$ | $\mathcal{L}_{PPD}$ | $\mathcal{L}_{KD}$ | OBQA | SIQA | ARC-c | WebQs |
|---|---|---|---|---|---|---|---|
| 1 | ✓ | ✓ | ✓ | **35.40** | **51.54** | **40.70** | **23.82** |
| 2 | ✓ | ✓ | ✗ | 26.00 | 44.32 | 37.80 | 16.88 |
| 3 | ✓ | ✗ | ✓ | 27.20 | 44.52 | 38.65 | 7.87 |
| 4 | ✗ | ✓ | ✓ | 32.20 | 46.42 | 38.57 | 23.43 |

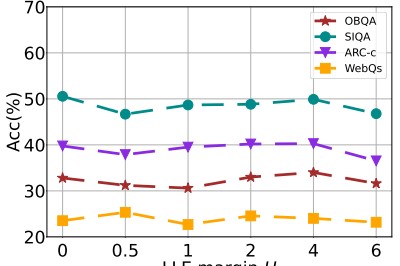 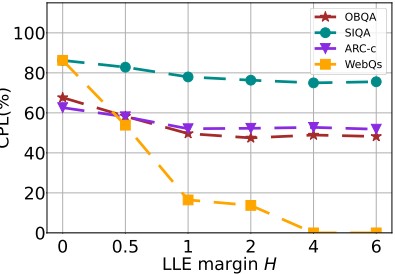

Figure 4: Details of the variation in average accuracy and CPL as the LLE margin $H$ increases across different tasks on Qwen2-1.5B-Instruct with the 0.5 dropout rate.

## 5.3 ABLATION STUDY

To systematically evaluate the effectiveness of each component of LLEOT, we conducted detailed ablation experiments. The experimental results are presented in Table 2, Table 3, and Figure 4, primarily focusing on three core components: the emulator construction phase, the knowledge distillation strategy, and the LLE margin $H$. To investigate the contributions of CPKD and LLE stages, we individually omitted them during the emulator construction process. When evaluating the knowledge distillation approach, we separately removed the language modeling loss (Equation 4), the proxy prompt distillation loss (Equation 3) and the knowledge distillation loss (Equation 2) from CPKD. Finally, for the ablation of $H$, we employed different $H$ values for LLE. Subsequently, we elucidate the role of each component individually.

**Impact of the Emulator Construction Phase.** As shown in rows 1 and 2 of Table 2, the absence of the CPKD stage during emulator construction leads to a significant decrease in the average accuracy of the original model after transfer. This indicates that CPKD effectively improves the applicability of the soft prompt, fine-tuned by the emulator, on the original model. Furthermore, a comparative analysis between rows 1 and 3 reveals that the absence of the LLE stage during emulator construction results in a decline in the model capability privacy protection metric, confirming LLE's effectiveness in preventing model capability privacy leakage. Additionally, we observed that when the dropout rate was 0.2, the decrease in average accuracy was not pronounced; however, when the dropout rate increased to 0.5, the average accuracy decreased significantly. This suggests that for emulators with lower compression rates, the application of CPKD may be optional.

**Impact of the Knowledge Distillation Strategy.** As shown in Table 3, removing any of the three loss terms ( $\mathcal{L}_{LM}$, $\mathcal{L}_{PPD}$, or $\mathcal{L}_{KD}$) results in a degradation of average accuracy, which confirms the necessity of combining all three in Equation 5. Notably, the largest performance degradation is observed upon the removal of $\mathcal{L}_{PPD}$, highlighting that our proposed $\mathcal{L}_{PPD}$ effectively enhances the applicability of soft prompts fine-tuned on the emulator to the original model.

**Impact of LLE margin.** Figure 4 reveals two key trends regarding the elevation margin $H$. First, the downstream accuracy remains remarkably robust to increases in $H$. This suggests that loss landscape geometric alignment, rather than the absolute loss value, is the critical factor for ensuring effectiveness of the fine-tuned soft prompt when applied to the original model. Second, the CPL

---

[1]https://github.com/EleutherAI/lm-evaluation-harness

metric decreases significantly for $H$ between 0 and 2, after which it plateaus. This indicates that the emulator's zero-shot performance converges to a lower bound with larger $H$. Consequently, this demonstrates that optimal capability privacy can be achieved without resorting to an overly large $H$.

# 6 CONCLUSION

In this work, we identify for the first time that existing OT methods carry the risk of model capability privacy leakage. To address this issue, we propose LLEOT, an innovative OT framework, whose core lies in the proposed LLE technique. We prove that this technique effectively disrupts the inference capability of emulators to prevent privacy leakage, while maintaining gradient consistency between the emulator and the original model. This ensures that adapters trained on the emulator remain applicable to the original model. Comprehensive experiments show that LLEOT achieves state-of-the-art performance in both protecting model privacy and model utility.

# 7 ETHICS STATEMENT

The research presented in this paper is fundamentally motivated by the ethical imperative to address significant privacy and security challenges in large model adaptation. Our work focuses on the Off-site Tuning paradigm, where a key ethical risk is the potential misuse of emulators that inadvertently leak the original model's inference capabilities. Our proposed method, LLEOT, is designed with a 'privacy-by-design' approach. The core Loss Landscape Elevation mechanism is intentionally engineered to degrade the emulator's inference abilities, thereby directly mitigating this risk of misuse. This work did not involve human participants or user studies. The methods and findings are intended solely for the research purpose of developing more secure, responsible, and trustworthy AI frameworks.

# 8 REPRODUCIBILITY STATEMENT

We have made every effort to ensure the reproducibility of our research. Our Loss Landscape Elevation Offsite Tuning (LLEOT) framework is detailed in Section 4, and its core mechanisms are formalized with pseudocode in Algorithm 1. All implementation details, including the base model architecture, hyperparameters for the LLE and CPKD phases, and the final prompt tuning setup, are provided in Appendix A.4. We conducted all experiments on publicly available academic benchmarks, including OpenBookQA, SocialIQA, ARC-Challenge, and WebQuestions. Specific details about the datasets are described in Appendix . To facilitate direct replication and further research, we will release our source code and emulator checkpoints upon publication, contributing to the open-source community.

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

## A  MORE DETAILS OF OUR METHOD

### A.1  LAYERDROP

The pseudocode of the LayerDrop algorithm is shown below.

---

**Algorithm 2** LayerDrop

---

**Input:** Original model $\mathcal{M}$, dropout rate $\beta$
**Output:** a list of layers
1: Get the layers of model: layers $\leftarrow |\mathcal{M}|$
2: $m, k \leftarrow$ len(layers), $\lfloor$len(layers) $\times \beta\rfloor$
3: stride $\leftarrow (m-1)/(k-1)$
4: **for** $j \leftarrow 0$ to $k-1$ **do**
5:     $i_j \leftarrow \lfloor j \times$ stride$\rfloor$
6: **end for**
7: **return** layers$[i_0, \ldots, i_{k-1}]$

---

### A.2  MODEL DETAILS

We conducted experiments on three commonly used LLMs: Qwen2-1.5B-Instruct[2], Gemma-2-2b-it[3], and Llama-3.2-3B-Instruct[4]. The architectural hyperparameters, training data size, and vocabulary size of these models are detailed as Table 4.

Table 4: Details of large language models.

| Models | Qwen2-1.5B-Instruct | Gemma-2-2b-it | Llama-3.2-3B-Instruct |
|---|---|---|---|
| Hidden Size | 1,536 | 2,304 | 3,072 |
| Layers | 28 | 26 | 28 |
| Query Heads | 12 | 8 | 24 |
| Key Value Heads | 2 | 4 | 8 |
| Head Size | 128 | 256 | 128 |
| Vocabulary Size | 151,936 | 256,000 | 128,256 |
| Trained Tokens | 7T | 2T | 9T |

### A.3  DATASET DETAILS

Table 5 summarizes the statistics of the downstream task datasets, while their corresponding instruction formats are presented in Tables 8.

Table 5: The statistics of downstream task datasets.

| Datasets | OBQA | SIQA | ARC-c | WebQs |
|---|---|---|---|---|
| Train Data Num | 5.0K | 33.4K | 2.3K | 3.8K |
| Test Data Num | 500 | 1,954 | 1,172 | 2,032 |
| Answer | Option | Option | Option | Option |

### A.4  IMPLEMENTATION DETAILS

In the CPKD stage, the emulator is distilled for one epoch on the initial $12.5\%$ of the first Pile-uncopyright chunk with a learning rate of $4e-6$; the loss weights ( $w_1$, $w_2$, $w_3$ ) are set to 1, 10, and 30, respectively. For the LLE stage, we experiment with two learning rates, $1e-6$ and $2e-6$,

---

[2]https://huggingface.co/Qwen/Qwen2-1.5B-Instruct
[3]https://huggingface.co/google/gemma-2-2b-it
[4]https://huggingface.co/meta-llama/Llama-3.2-3B-Instruct

Table 6: Results of ablation experiments on loss landscape elevation methods. NLM denotes elevation using negative language modeling loss. Best in **bold**.

| DR | Method | OBQA | | SIQA | | ARC-c | | WebQs | |
|---|---|---|---|---|---|---|---|---|---|
| | | Acc($\uparrow$) | CPL($\downarrow$) | Acc($\uparrow$) | CPL($\downarrow$) | Acc($\uparrow$) | CPL($\downarrow$) | Acc($\uparrow$) | CPL($\downarrow$) |
| 0.2 | NLM | 24.80 | 64.03 | 49.90 | **69.94** | 20.56 | **59.62** | 13.29 | 0.00 |
| | LLE | **33.87** | **49.40** | **53.34** | 75.70 | **41.98** | 61.00 | **28.75** | **0.00** |
| 0.5 | NLM | 23.00 | 66.19 | 42.68 | **69.83** | 20.03 | 63.52 | 5.86 | 0.00 |
| | LLE | **34.20** | **49.88** | **50.04** | 76.31 | **40.44** | **49.30** | **24.15** | **0.00** |

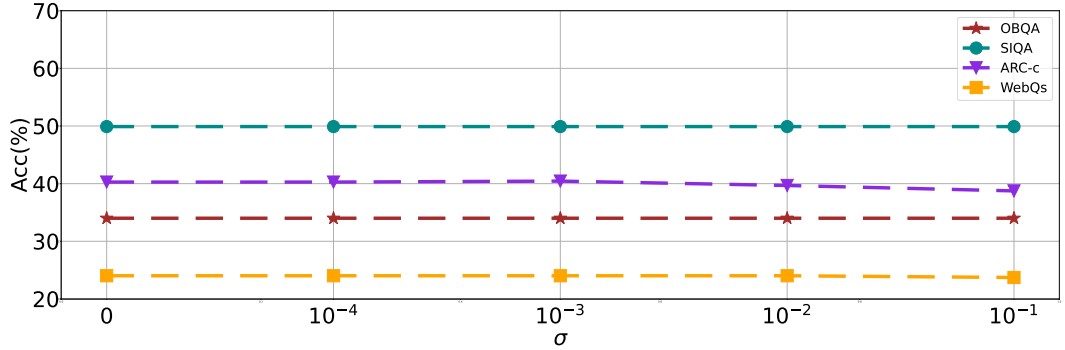

Figure 5: Variations in the average accuracy of LLEOT across different noise intensities on Qwen2-1.5B (dropout rate = 0.5). Here, $\sigma$ denotes the standard deviation of Gaussian noise applied to the fine-tuned soft prompts.

using the initial $1\%$ of the same data chunk. We select and report the results from the emulator that achieves the best performance. In the comparative experiments, the elevation margin $H$ is set to 4. During downstream fine-tuning, we use a soft prompt of length 5 and conduct a grid search over learning rates, reporting the best-performing run. The search grids are $\{1e-1, 7e-2, 3e-2, 3e-3, 1e-3\}$ for Qwen2-1.5B-Instruct and Gemma-2-2b-it, and $\{5e-3, 1e-3, 5e-4, 1e-4, 5e-5\}$ for Llama-3.2-3B-Instruct. All experiments were conducted on two NVIDIA A800 GPUs.

## B ADDITIONAL EXPERIMENTS AND ANALYSIS

### B.1 ABLATION EXPERIMENTS ON LOSS LANDSCAPE ELEVATION METHODS

We conduct an ablation study on the method of loss landscape elevation to demonstrate that our proposed strategy effectively preserves gradient similarity between the elevated emulator and the original model during prompt tuning. For comparison, we use an unconstrained elevation method as a baseline. Specifically, this baseline directly elevates the emulator's loss landscape using the negative language model loss. The results are presented in Table 6.

As shown in Table 6, compared to the unconstrained elevation method, our proposed approach significantly improves the final accuracy while effectively protecting model capability privacy. This strongly indicates that our elevation strategy successfully maintains gradient similarity between the emulator and the original model during prompt tuning, thereby enabling the fine-tuned soft prompts to be highly applicable to the original model.

### B.2 COMPATIBILITY WITH DATA PRIVACY STRATEGIES

To verify that the LLEOT framework is orthogonal to data privacy strategies, we incorporate the widely used randomization privacy protection strategy (Zhu et al., 2019; Kang et al., 2023) into

LLEOT. Specifically, after prompt tuning, the data owner adds Gaussian noise to the soft prompt before uploading it to the model owner. This method is known to significantly reduce the success rate of gradient inversion attacks (Zhu et al., 2019), thereby preventing the model owner from deducing private data.

Figure 5 illustrates the variation in the original model's average accuracy with the introduction of noise intensity. Unexpectedly, the noise added to the soft prompts has a negligible impact on model performance. We attribute that this robustness stems from the high smoothness of the original model's input embedding space, resulting from its pre-training on massive amounts of data. This smoothness ensures that small perturbations to the embedding vectors do not significantly alter the model's output. Therefore, the randomization privacy protection strategy can be integrated into the LLEOT framework, enhancing data privacy at a negligible cost to performance.

### B.3 COMPARISON OF ADAPTER SIZES

As shown in Table 7, our method employs an adapter with a parameter count that is significantly lower than the existing methods, thereby drastically reducing the consumption of computational resources.

Table 7: Parameter counts of adapters for different methods.

| Method | OT | CRaSh | Ours |
|---|---|---|---|
| Qwen2-1.5B-Instruct | 187.2M | 187.2M | 7.6K |
| Gemma-2-2b-it | 311.5M | 311.5M | 11.5K |
| Llama-3.2-3B-Instruct | 402.7M | 402.7M | 15.4K |

## C THE USE OF LLMS

In the preparation of this manuscript, we employed a large language model (LLM), specifically Gemini 2.5 Pro (Comanici et al., 2025), as a writing aid. The LLM's role was explicitly restricted to language refinement and did not involve any facet of the research conceptualization or scientific methodology. Our process consisted of providing the LLM with drafts and specific sentences. We then utilized the model's suggestions to polish sentence construction, enhance clarity and flow, and verify grammatical accuracy in the final text. It is essential to declare that all central scientific contributions—including the motivation for this study, the definition of the model capability privacy concept and its associated metric, the algorithmic architecture and theoretical analysis of LLEOT, and the experimental design and interpretation of results—are exclusively the work of the human authors. The LLM was not utilized to formulate scientific claims, hypotheses, or conclusions. In compliance with ICLR policy, the authors have fastidiously reviewed, edited, and confirmed all content in this paper. We assume complete responsibility for the final manuscript, encompassing its scientific precision and integrity.

## D PROOF OF THEOREM 1.

*Proof.* We first show that LLE effectively degrades the emulator's inference ability. From the definition of cross-entropy loss and Eq. 6, we obtain

$$\mathcal{L}_{\mathcal{E}}(P; x; y) - \mathcal{L}_{\mathcal{M}}(P; x; y)$$

$$= -\tfrac{1}{n} \sum_{i=1}^{n} \log(p_{\mathcal{E}}(x_i \mid P, x_{1:i-1})) + \tfrac{1}{n} \sum_{i=1}^{n} \log(p_{\mathcal{M}}(x_i \mid P, x_{1:i-1}))$$

$$= H > 0. \tag{12}$$

Here, $n$ denotes the number of tokens to predict in $x$, $p_{\mathcal{E}}(x_i|P, x_{1:i-1})$ denotes the probability of the emulator correctly predicting the $i - th$ token given the soft prompt and the preceding $i - 1$ tokens. Defining $\hat{p}_{\mathcal{E}}(x|P) = \prod_{i=1}^{n} p_{\mathcal{E}}(x_i|P, x_{1:i-1})$ and $\hat{p}_{\mathcal{M}}(x|P) = \prod_{i=1}^{n} p_{\mathcal{M}}(x_i|P, x_{1:i-1})$, Equation 12

can be transformed into:

$$\hat{p}_{\mathcal{E}}(x|P) = \mathrm{e}^{-nH}\hat{p}_{\mathcal{M}}(x|P) \tag{13}$$

To analyze the impact of this loss difference on model performance, we consider the perplexity (PPL), a standard metric for evaluating language models. Perplexity is defined as:

$$\mathrm{PPL} = \exp\left(-\frac{1}{n}\sum_{i=1}^{n}\log(p_i)\right) = \exp\left(-\frac{1}{n}\log(\hat{p})\right) = \hat{p}^{-1/n}. \tag{14}$$

From Equation 14, the perplexity of the emulator can be expressed as:

$$\begin{aligned}
\mathrm{PPL}_{\mathcal{E}} &= \exp\left(-\frac{1}{n}\sum_{i=1}^{n}\log(p_{\mathcal{E}}(x_i \mid P, x_{1:i-1}))\right) \\
&= \exp\left(-\frac{1}{n}\log(\prod_{i=1}^{n}p_{\mathcal{E}}(x_i \mid P, x_{1:i-1}))\right) \\
&= \exp\left(-\frac{1}{n}\log(\hat{p}_{\mathcal{E}}(x|P))\right) \\
&= \hat{p}_{\mathcal{E}}(x|P)^{-1/n}.
\end{aligned} \tag{15}$$

This can be rewritten as:

$$\hat{p}_{\mathcal{E}}(x|P) = \mathrm{PPL}_{\mathcal{E}}^{-n}. \tag{16}$$

Similarly, for the original model, we have:

$$\hat{p}_{\mathcal{M}}(x|P) = \mathrm{PPL}_{\mathcal{M}}^{-n}. \tag{17}$$

By substituting Equation 16 and Equation 17 into Equation 13, we can express the relationship between the perplexities of the two models:

$$\mathrm{PPL}_{\mathcal{E}} = \mathrm{e}^{H} \cdot \mathrm{PPL}_{\mathcal{M}}. \tag{18}$$

It shows that the emulator's PPL is exponentially greater than the original model's by a factor of $\mathrm{e}^{H}$. Since lower PPL indicates better performance, a larger $H$ will lead to a significantly higher PPL for the emulator, thereby degrading its inference capabilities.

In addition, we show that LLE maintains the emulator's gradient guidance consistent with that of the original model. Specifically, the emulator's gradient with respect to soft prompts can be expressed as:

$$\nabla_P \mathcal{L}_{\mathcal{E}}(P; x; y) = \nabla_P\left(\mathcal{L}_M(P; x; y) + H\right) = \nabla_P \mathcal{L}_M(P; x; y) + \nabla_P H = \nabla_P \mathcal{L}_M(P; x; y), \tag{19}$$

the gradient vectors of the emulator and the original model are identical. During prompt tuning, the emulator and the original model exhibit consistent gradient optimization directions and magnitudes at each step, ultimately converging to the same optimal soft prompt. $\qquad\square$

Table 8: Instructions format of downstream task dataset

| *OBQA* |
| --- |
| What happens when mercury is placed in water? 
 *it sinks.* |
| Which is a good source of nutrients for a mushroom? 
 *a cut peony.* |

| *SIQA* |
| --- |
| Q: Sydney was a school teacher and made sure their students learned well. How would you describe Sydney? 
 A: 
 *As someone that takes teaching seriously.* |
| Q: Kendall's dog was overweight so they walked it five miles. Why did Kendall do this? 
 A: 
 *start an exercise regimen.* |

| *ARC-c* |
| --- |
| Question: What do cells break down to produce energy? 
 Answer: 
 *food.* |
| Question: How are the particles in a block of iron affected when the block is melted? 
 Answer: 
 *The particles move more rapidly.* |

| *WebQs* |
| --- |
| Question: what is nina dobrev nationality? 
 Answer: 
 *Bulgaria.* |
| Question: what electorate does anna bligh represent? 
 Answer: 
 *Electoral district of South Brisbane.* |

