# OpenReview forum: "LLEOT: A Privacy-Enhancing Offsite Tuning Framework via Loss Landscape Elevation"
_ICLR.cc/2026/Conference — ICLR 2026 Conference Withdrawn Submission_

### Official Review · Reviewer_SphT · 2025-10-21

**Soundness:** 2
**Presentation:** 2
**Contribution:** 3
**Rating:** 4
**Confidence:** 3

**Summary:**

This paper proposes LLEOT a framework that secures both data and model capability privacy.  Its core component, Loss Landscape Elevation (LLE), enforces a fixed loss margin between an emulator of the system and the original model.  The authors theoretically show that LLE (i) degrades emulator inference through perplexity amplification and (ii) preserves gradient alignment, ensuring consistent convergence of prompt optimization.  The authors conduct extensive experiments on various question-and-answering datasets to confirm LLEOT achieves strong adaptation while mitigating emulator noise.

**Strengths:**

The paper addresses an interesting and practical challenge of fine-tuning pre-trained LLMs in a setting where, both, the model and data are private and cannot be disclosed.  The proposed solution allows the model maintainer to train a surrogate model, that preserves model capability privacy, and to distribute this surrogate to the data maintainer for refinement.  The paper is largely well-written and the methodology appears novel.

**Weaknesses:**

Critically, this paper does not discuss how the parameter of the emulator are adapted back into the original model.  The paper states, "We require that transferring the trained weights $\Delta^*$ back to the original model...should yield performance comparable to that achieved by directly optimizing $\mathcal{M}$ on the dataset...without requiring access to $\mathcal{M}$ itself."  This seems like a strong constraint that is nontrivial to enforce and it is unclear how the authors ensure this parameter transfer remains valid.  This is particularly concerning given that the emulator is intentionally degraded so as to not leak model capability.

Another critical issue is that it is somewhat mysterious how the authors enforce the LLE constraint in Eq. (6)--note that $\mathcal{L}_{\mathcal{M}}$ is never defined.  Ensuring that the loss landscape between the emulator and model are equivalent, up to an additive constant, seems like a nontrivial constraint to enforce, yet the authors provide no discussion of how this is done in practice.  Indeed, Algorithm 1 (Line 12) states "Optimize $\mathcal{E}^*$ with respect to Equation (10)" which I suspect is a typo as Eq. (10) is not related to the LLE optimization at all.  Looking over the proof of Theorem (1) it is also not obvious how the authors go from Eq. (12) to Eq. (14) as this is not a simple exponentiation of Eq. (12) (the constant factor is incorrect).

A slightly lower priority issue is that the evaluation methodology needs some clarification.  In particular it is not clear whether the accuracy numbers reported are for the emulator (after fine tuning) or if they are accuracy on the original model $\mathcal{M}$ after adaptation of the new parameters (i.e. $\mathcal{M}_{\Theta + \Delta^*}$).

Some minor issues:
* Algorithm 1 (Line 19) : I think the gradient update is incorrect here as $P^*$ is never initialized to anything
* L291-292 : The statement is incorrect, I think you need to swap the references to the model and the emulator
* L374 : Change "upper bound" to "lower bound"

**Questions:**

Theorem 1 states that the perplexity of the emulator scales exponentially with perplexity of the original model.  One would expect that, as H increases, emulator performance would degrade exponentially.  Yet Fig. 4 suggests this is not the case, and in fact there is relatively little relationship between H and the model inference capability.  Please clarify this apparent contradiction.

---

> ### Author Response · Authors · 2025-11-21
>
> We appreciate your acknowledgment of the contribution and innovation of our work. Below, we address your questions and suggestions point by point. All line, table and figure numbers refer to the latest PDF.
>
> # 1. How the emulator parameters are adapted back to the original model and how the effectiveness of parameter transfer is guaranteed (w1):
> **We only transfer the adapter trained by the emulator (L319), not the emulator's parameters.**
>
> The complete workflow of LLEOT is as follows:
>
> 1.**Emulator Construction**:After the model owner constructs the emulator through layerdrop, CPKD and LLE, it sends the emulator to the data owner.
>
> 2.**Adapter Training**:The data owner performs prompt tuning on the emulator using local data and sends the trained soft prompt back to the model owner.
>
> 3.**Adapter Transfer**:The model owner integrates the received soft prompt into the original model, thereby enabling it to acquire the knowledge from the local data.
>
> Therefore, $ \Delta^* $ represents the adapter weights (which are soft prompts $P^*$ in this paper).
>
> We ensure the effectiveness of this parameter transfer through LLE, which aligns the geometry of the loss landscapes of the emulator and the original model. This alignment **ensures that their gradients with respect to the adapter are consistent**, thereby making the adapter trained by the emulator applicable to the original model.
>
> # 2. How the LLE constraint is executed and the derivation of Equations (12) to (14) (w2):
>
> **In the LLE phase, we actually optimize the emulator with respect to Eq. (6)**, not Eq. (10) (a typo). In Eq. (6), $\mathcal{L}\_{\mathcal{M}}$ denotes the prompt tuning loss for the original model (L275). For clarity, in the revised version, we add a specific optimization objective:
>
> $$
> \mathcal{E}^{*}=\mathop{\arg\min}\limits_{\mathcal{E}}\mathbb{E}_{x\sim\mathcal{X}_e ,P'\sim \mathcal{N}(\mu, \sigma^2)} \left| \mathcal{L}\_{ \mathcal{E} }(P'; x) - \mathcal{L}\_{\mathcal{M}}(P'; x) - H \right|
> $$
>
> In the original paper's derivation from Eq. (12) to Eq. (14), the statement "By exponentiating both sides of Equation 12" may be overly concise. We now present a more complete derivation to clarify the process:
> From the definition of perplexity in Eq. (13), we have:
> $$
> \text{PPL}\_\mathcal{E} = \exp\left( -\frac{1}{n} \sum\_{i=1}^{n} \log(p_{\mathcal{E}}(x\_{i} \mid P,x\_{1:i-1})) \right)
> = \exp\left( -\frac{1}{n} \log(\prod\_{i=1}^{n}p_{\mathcal{E}}(x\_{i} \mid P,x\_{1:i-1})) \right)
> = \exp\left( -\frac{1}{n} \log(\hat{p}\_{\mathcal{E}}(x|P)) \right)
> = \hat{p}\_{\mathcal{E}}(x|P)^{-1/n}
> $$
>
> $$
> \hat{p}\_{\mathcal{E}}(x|P) = \text{PPL}\_\mathcal{E}^{-n}
> $$
>
> Similarly:
> $$
> \hat{p}\_{\mathcal{M}}(x|P) = \text{PPL}\_\mathcal{M}^{-n}
> $$
>
> Substituting the above expressions into Eq. (12), we obtain Eq. (14):
> $$
> \text{PPL}\_\mathcal{E} = \mathrm{e}^{H} \cdot \text{PPL}\_\mathcal{M},
> $$
>
> # 3. The specific meaning of the reported accuracy (w3):
> In Table 1 of the main text, the "Acc" is defined as follows:
>
> 1.For Full ZS, it is the zero-shot accuracy of the original model.
>
> 2.For Full PT, it is the accuracy of the original model after being prompt-tuned.
>
> 3.For Random, it is the zero-shot accuracy of a randomly initialized model.
>
> 4.For OT, Crash and Ours, it is the accuracy of the original model after incorporating the adapter trained on the emulator (i.e. $\mathcal{M}\_{ \Theta+\Delta^* }$).
>
> Furthermore, in Tables 2, 3, and Figure 4, the "Acc" value also represents the accuracy of the original model after incorporating the adapter trained on the emulator.
>
> # 4. Inappropriate expressions (w4):
> Thank you for your suggestion.
>
> Algorithm 1(Line20): We have updated the manuscript to clarify that the soft prompt is initialized using the corresponding word embeddings.
>
> L297-298: This has been corrected in the revised manuscript.
>
> L377: "upper bound" is correct, as a randomly initialized model's minimal zero-shot performance corresponds to the least leakage of the original model's capabilities, thereby providing the strongest privacy protection.
>
> # 5. Why the emulator's performance does not decrease exponentially as $H$ increases (q1):
> As shown in Table 1, the perplexity of the emulator constructed by our method increases exponentially as $H$ increases. **The reason why the emulator's performance does not decrease exponentially in tandem is that the relationship between perplexity and model performance is not linear.** As $H$ increases, the emulator's performance does not decrease exponentially but rather approaches a lower bound, which is close to the performance of a randomly initialized model.
>
> **Table 1: Theoretical vs. empirical PPL of the emulator under different $H$ (Qwen2-1.5b, DR=0.5). Following OT, we use wikitext to validate the PPL. The results show that the emulator's PPL aligns with theoretical expectations.**
> |Dataset\H|2|4|6|8|
> |:-:|:-:|:-:|:-:|:-:|
> |Empirical PPL|136.44|758.17|4782.65|37940.29|
> |Theoretical PPL|98.79|729.94|5393.60|39853.58|

---

> > ### Comment · Reviewer_SphT · 2025-11-21
> > **Thank you for your responses**
> >
> > Thank you for your prompt reply to my comments.  Below are my responses:
> >
> > 1: Unfortunately I am unable to follow this explanation without a thorough reread of the paper.  I highly recommend revising the manuscript to clarify the workflow.
> >
> > 2: Thank you, this math checks out!  I recommend putting this derivation into the appendix.
> >
> > 3: This really needs to be clarified in the manuscript.
> >
> > 4: It sounds like you have already revised to include these changes.
> >
> > 5: Thanks for the explanation, this makes sense.
> >
> > I'm afraid that I am of the opinion that the manuscript as-is is not in a publishable form.  I recommend incorporating feedback and making the necessary revisions for clarification.  I will stick with my current score.

---

> ### Author Response · Authors · 2025-11-22
>
> We sincerely appreciate the time you dedicated to reviewing our manuscript and your thoughtful feedback. In the revised manuscript, we have implemented the following modifications:
>
> (1) We have added a detailed description of the workflow to the Setup subsection of the Problem Formulation (L139-151) and the opening paragraph of the Methodology section (L180-189).
>
> (2) We have clarified the specific definition of the reported accuracy in the Metrics subsection of the Experiments section (L402-406).
>
> (3) We have provided the detailed proof of Theorem 1 in the Appendix (L873-890).
>
> (4) We have corrected the inappropriate expressions as suggested.
>
> Once again, we thank you for your valuable suggestions.

---

### Official Review · Reviewer_rGBy · 2025-10-28

**Soundness:** 2
**Presentation:** 2
**Contribution:** 1
**Rating:** 2
**Confidence:** 4

**Summary:**

This paper introduces an intermediate model to isolate user data from direct access to the closed-source model. By combining knowledge distillation and local fine-tuning strategies, a new model is ultimately trained to adapt to the user’s data. The paper also provides theoretical analyses to support the feasibility of convergence and conducts experimental validation on several standard datasets.

**Strengths:**

1. This paper focuses on privacy scenarios involving data and closed-source models, demonstrating a certain degree of practical foresight and forward-looking applicability.

2. The paper presents a complete and coherent narrative, clearly describing the entire training process. The language is well-organized and easy to understand.

**Weaknesses:**

1. The algorithmic design in the paper does not strictly adhere to its initial idea; in fact, the subsequent experimental section exhibits significant shortcomings. The original design intention (line 144), the architecture and parameters of this emulator should differ from those of the original model, and its inference capability should approximate that of a randomly initialized model. However, in the later algorithmic implementation, the authors construct the emulator by applying layer pruning to the original model, effectively assembling a subnetwork that retains portions of the original model connected through skip connections. This approach contradicts the stated objective, as the resulting emulator is not an independent model but rather a structurally reduced version of the original one.

2. Such simplification is unacceptable, as it would inevitably lead to the leakage of the original parameters of the closed-source model. In the algorithm section, the authors do not mention whether noise injection or other privacy-preserving mechanisms are applied to the pruned parameters. As a result, the current version of the work presents serious issues in terms of experimental validity and privacy protection.

3. The formulation in Equation (6) does not appear to generalize to cases where the emulator is initialized as a random model. In essence, the effectiveness of LLE relies on the assumption that the loss landscapes of the model $M$ and $\theta$ are sufficiently similar. This requirement implicitly necessitates a degree of parameter correlation or alignment between the two models—an assumption that reintroduces the parameter exposure problem. Consequently, the claimed privacy preservation objective is in conflict with the underlying design of LLE.

4. There is a lack of genuine privacy analysis in the paper. The authors do not provide any effective quantitative evaluation of how much improvement is achieved in terms of model parameter privacy and data privacy, respectively. Without explicit metrics or empirical validation, the claimed privacy enhancement remains qualitative and unsubstantiated.

5. There is no effective convergence analysis provided in the paper. Although the LLE formulation ensures that the emulator’s gradient maintains the same direction as the original gradient, it remains unclear whether incorporating the soft prompt feature alignment loss introduces gradient distortion during the optimization of the original text-based training process. In other words, the paper does not analyze whether the combined objectives preserve the global convergence properties or potentially alter the optimization dynamics of the original model.

**Questions:**

1. Line 144 mentions that the proxy model should exhibit performance comparable to that of the original model under random parameter initialization. Could the authors clarify the Weak 1 and 2? Specifically, what protective measures are implemented to safeguard the original parameters when layer pruning is used in the experiments? Moreover, is it possible that, through multiple query accesses, one could reconstruct or steal the full set of model parameters from the pruned model?

2. What is the theoretical basis for the term $H$ in Equation (6), and why can it be considered equal?

3. Why does CPKD show completely irregular performance across different pruning ratios in Table 2? When the pruning ratio is 0.2, CPKD has almost no effect, but at 0.5, it has a significant impact. What causes this phenomenon?

4. Why does WebQA perform so poorly under the third loss combination in Table 3 (much lower than the other three datasets)?

5. How were the CPL results in the paper evaluated? I couldn’t find a detailed explanation of the testing procedure like which dataset were the CPL measurements conducted, and what specific tests were performed?

6. Why do some of the training results for the pruned model in Table 1 exceed those of the full-parameter fine-tuning model? This seems unreasonable. Does it imply that the results of the full-parameter fine-tuning are not actually optimal?

7. How should the length of the soft prompt be selected during the training phase in the experiments?

---

> ### Author Response · Authors · 2025-11-21
>
> We appreciate your insightful suggestions. Below, we address your concerns in detail. All line, table and figure numbers refer to the latest PDF.
>
> # 1. The potential leakage of the original model's parameters and the possibility of reconstructing or stealing the full model through multiple queries (w1, w2, w3, q1):
> In our method, **the emulator's parameters are altered during the CPKD and LLE phases, ensuring they differ from the original model's and thus preventing parameter leakage**. Table 1 reports the sum of the L1 and L2 norm distances between the emulator's parameters and the corresponding original model parameters. In contrast, methods like Crash, which construct the emulator solely through pruning, do cause the emulator to retain a subset of the original model's parameters.
>
> **Table 1: The sum of the L1 and L2 norm distances. (Qwen2-1.5b, DR=0.5)**
> |L1|L2|
> |:-:|:-:|
> |2907146.49|150.78|
>
> Furthermore, **it is infeasible to steal the original model's parameters by multiple querying the emulator built by our method**. This is because our approach significantly increases the emulator's perplexity during the LLE phase. Consequently, the emulator exhibits extremely poor zero-shot performance (as indicated by the CPL metric in Table 1 of the main text), and its outputs are meaningless. As such, this output cannot be exploited to steal the original model's parameters.
>
> # 3. The lack of analysis for model parameter privacy and data privacy (w4):
> **Our method protects model parameter privacy** because our emulator's parameters diverge from the original model's (as shown in Table 1), thus preserving model parameter privacy when the emulator is sent to the data owner. **Our method also protects data privacy**, as it is orthogonal to data privacy techniques like differential privacy (as shown in Appendix B.2), ensuring that the transfer of the soft prompt doesn't leak user data.
>
> Furthermore, **our claimed privacy enhancement focuses on further protecting model capability privacy**, building upon existing methods that already safeguard model parameter and data privacy. Our contribution is not centered on enhancing protections for model parameters or data.
>
> # 4. Does the soft prompt feature alignment loss alter the original model's optimization dynamics (w5):
> **The Proxy Prompt Distillation Loss introduced in CPKD only impacts the emulator and doesn't alter the optimization dynamics of the original model.** During CPKD phase, all parameters of the original model are frozen.
>
> # 5. The justification for Eq. (6) (q2):
> **Eq. (6) serves as the optimization objective for the LLE phase, rather than a theorem that holds true under all conditions**. During the LLE stage, we optimize the emulator such that the difference in prompt tuning loss between the emulator and the original model approximates $H$ for any given input.
>
> # 6. Why CPKD has a different impact at varying DR settings (q3):
> **When DR=0.2, the difference between the feature representations of the emulator and the original model is smaller than when DR=0.5 (as shown in Table 2)**. Therefore, the effect of CPKD is less significant in the DR=0.2 case.
>
> **Table 2: Average L2 distance (Qwen2-1.5b).**
> |DR|OBQA|SIQA|ARC-c|WebQs|
> |:-:|:-:|:-:|:-:|:-:|
> |0.2|0.68|0.70|0.72|0.76|
> |0.5|0.95|1.01|1.02|0.99|
>
> # 7. The low WebQs score for the third loss combination in Table 3 (q4):
> The poor performance on WebQs is due to its high difficulty (Qwen2-1.5b's zero-shot score is only 1.82), **making the results highly sensitive to the quality of the soft prompt**. Since the soft prompt produced by the emulator in this configuration was of lower quality, the WebQs score dropped significantly.
>
> # 8. CPL Evaluation (q5):
> In tables of the main text, the datasets listed in the column headers (e.g., OBQA) are the evaluation datasets for the CPL metric. the CPL metric is the ratio of the zero-shot score of the emulator to that of the original model on a given dataset (L152-161). A lower CPL score indicates stronger protection for model capability privacy.
>
> # 9. Why some pruned models in Table 1 outperform the full-parameter fine-tuned model (q6):
> The Full PT results in Table 1 of the main text are from prompt tuning, not full-parameter fine-tuning. We speculate that the reason the adapter trained by the emulator slightly exceeds the Full PT is that **direct prompt tuning on the original model may lead to a degree of overfitting**. This outcome further demonstrates the robustness of our method.
>
> # 10. The selection of the prompt length (q7):
> We set the prompt length to 5. This choice is motivated by the fact that it yields the most lightweight adapter, and our prompt tuning experiments on the original model show that performance is stable across different length settings (as shown in Table 3).
>
> **Table 3: Acc with different prompt lengths (Qwen2-1.5b).**
> |Dataset\prompt length|5|10|20|50|
> |:-:|:-:|:-:|:-:|:-:|
> |OBQA|35.80|34.80|35.80|35.20|
> |ARC-c|41.98|41.98|42.49|43.17|

---

> > ### Comment · Reviewer_rGBy · 2025-11-26
> > **Thanks for the responses**
> >
> > Thank you for the authors’ response. The reply appears rather brief and lacks sufficient evidence to address the issues raised. For example, the authors claim that the emulator parameters differ from those of the original model, yet in the experiments the emulator parameters are initialized by pruning from the original model. This initialization process alone inevitably leads to parameter leakage. I believe this point has not been adequately clarified.
> >
> > The authors attribute the ineffective results on certain datasets to the presumed difficulty of the tasks, without conducting additional investigations or providing empirical evidence. This explanation lacks credibility. Moreover, I suggest renaming “full finetune,” as it is clearly misleading and conflicts with the common meaning of full-parameter fine-tuning.
> >
> > In addition, the authors should provide the complete hyperparameter search configurations for all experiments, including those of the baselines. The claim that some baseline results might suffer from overfitting does not demonstrate the robustness of the proposed method. Instead, it indicates that the baseline experiments themselves are unreliable. At the very least, the authors should provide non-overfitted baseline results for fair comparison; otherwise, the reported performance gains on downstream fine-tuning tasks lack practical significance.

---

### Official Review · Reviewer_mgAz · 2025-11-02

**Soundness:** 2
**Presentation:** 3
**Contribution:** 3
**Rating:** 4
**Confidence:** 4

**Summary:**

This study is the first to reveal that existing Optimal Transport (OT) methods may inadvertently expose a model’s capability, posing privacy risks. To mitigate this, the authors introduce LLEOT, a novel OT framework centered around the Latent Leakage Elimination (LLE) technique. The proposed method effectively disrupts emulator inference to prevent privacy leakage while preserving gradient consistency between the emulator and the original model. This preservation ensures that adapters trained on the emulator remain valid when applied to the original model. Extensive experiments demonstrate that LLEOT achieves state-of-the-art performance in safeguarding model privacy without compromising model utility.

**Strengths:**

1. The study uncovers a previously neglected risk of model capability privacy leakage in Offsite Tuning, revealing that existing emulators can retain significant inference capability, potentially allowing malicious data owners to extract or exploit proprietary model knowledge.

2. The proposed Loss Landscape Elevation Offsite Tuning (LLEOT) framework introduces the Loss Landscape Elevation (LLE) technique, which effectively disables emulator inference while maintaining gradient alignment with the original model. Theoretical analysis (Theorem 1) guarantees that LLE increases emulator perplexity and preserves convergence toward the same optimal prompt.

3. The integration of LLE with Collaborative Prompt Knowledge Distillation (CPKD) enables efficient distillation for soft prompts, ensuring that adapters trained on the emulator transfer seamlessly to the original model.

4. Extensive experiments confirm that LLEOT delivers state-of-the-art results, achieving stronger privacy protection without sacrificing model performance compared to existing approaches.

**Weaknesses:**

1. Limited Baseline Comparison
The experimental evaluation includes only two baselines — Offsite Tuning (OT) and CRaSh (2023) — which are insufficient to represent the current state of the field. Considering the rapid progress in model adaptation and privacy-preserving fine-tuning methods in 2024–2025, incorporating more recent baselines is necessary for a fair and comprehensive comparison. Additionally, comparison with other privacy-oriented LLM training frameworks would further demonstrate the effectiveness and generality of the proposed approach.

2. Complex Training Procedure and Reproducibility Concerns
While the proposed method is promising, the training pipeline appears overly complex, involving multiple loss functions and hyperparameters (e.g., ω₁, ω₂, and ω₃ in Equation 5). The paper does not analyze the training stability or sensitivity to these parameters, which raises concerns about robustness. Without such analysis, the reproducibility and practicality of the method in real-world or large-scale scenarios remain uncertain.

**Questions:**

see Weaknesses

---

> ### Author Response · Authors · 2025-11-21
>
> We appreciate your acknowledgment of the novelty of our proposed approach. Below, we respond to your questions and concerns in detail. All line, table and figure numbers refer to the latest PDF.
>
> # 1. Supplemental Comparative Methods (w1):
>
> **In the subsequent version, we will include the latest state-of-the-art (SOTA) method, GradOT, as a comparative method.** We did not include this work in our initial submission because it is a closed-source paper published in August 2025, only one month prior to the ICLR submission deadline. We have since replicated GradOT and aligned its experimental setup with ours. Below, we present the partial results of the comparative experiments.
>
> As detailed in Table 1, our method surpasses GradOT in both average accuracy and the CPL metric. This indicates that our approach provides more effective adapter transfer and stronger protection for model capability privacy. GradOT's poor performance may be attributed to its sensitivity to hyperparameters, leading to suboptimal results.
>
> **Table 1: Comparative experimental results (DR=0.5). Best performance bolded.**
> |Method|OBQA|SIQA|ARC-c|WebQs|
> |:-:|:-:|:-:|:-:|:-:|
> |-|Acc(↑) CPL(↓)|Acc(↑) CPL(↓)|Acc(↑) CPL(↓)|Acc(↑) CPL(↓)|
> |||Qwen2-1.5b|||
> |GradOT|28.60 50.36|48.00 **74.33**|37.97 56.18|11.61 0.00|
> |Ours|**34.20 46.52**|**50.04** 75.87|**40.44 48.39**|**24.15 0.00**|
> |||Gemma2-2b|||
> |GradOT|38.40 43.26|50.46 **69.08**|50.63 42.50|18.20 0.00|
> |Ours|**44.87 38.01**|**55.01** 70.15|**52.56 39.76**|**22.44 0.00**|
> |||Llama3.2-3b|||
> |GradOT|29.80 55.32|39.66 **76.80**|45.05 49.21|10.77 0.00|
> |Ours|**33.40 53.43**|**48.21** 81.25|**46.96 47.19**|**15.45 0.00**|
>
> References:
>
> [1] GradOT: Training-free Gradient-preserving Offsite-tuning for Large Language Models. (Kai Yao, et al., ACL 2025).
>
>
> # 2. Supplemental Ablation Experiments (w2):
>
> **Our method's complexity is comparable to the baselines.** In comparison to OT, which comprises two stages (pruning and distillation), our method incorporates one additional phase: LLE. The LLE stage is simple to implement and does not meaningfully increase the method's overall complexity.
>
> Furthermore, **our method exhibits low sensitivity to hyperparameters and is highly robust**. To better demonstrate the robustness of our method, we conduct additional ablation studies on several key hyperparameters: the variance of the proxy soft prompt sampling distribution($\sigma$) and the CPKD loss weights (w2 and w3). Below, we present the results of these ablation experiments under the Qwen2-1.5B setting with DR=0.5.
> (Ablation studies for the key LLE hyperparameter, $H$, were already included in the main text (Figure 4). Consequently, our new ablation experiments focus exclusively on the hyperparameters related to CPKD. For this reason, the results presented here correspond to emulators trained only up to the CPKD phase, without the subsequent LLE phase.)
>
> As shown in Tables 2, 3, and 4, the performance of our method remains stable across different hyperparameter settings, demonstrating its low sensitivity to them.
>
> **Table 2: Results of ablation experiments for $\sigma$.**
> |Dataset\sigma|5|10|20|30|
> |:-:|:-:|:-:|:-:|:-:|
> |OBQA|32.20|31.20|35.40|29.20|
> |SIQA|50.92|50.46|51.54|49.03|
> |ARC-c|39.42|40.78|40.70|39.51|
> |WebQs|26.08|26.72|23.82|23.82|
>
> **Table 3: Results of ablation experiments for w2.**
> |Dataset\w2|0|10|20|30|
> |:-:|:-:|:-:|:-:|:-:|
> |OBQA|27.20|35.40|31.00|31.60|
> |SIQA|44.52|51.54|49.13|48.31|
> |ARC-c|38.65|40.70|39.51|39.16|
> |WebQs|7.87|23.82|25.94|24.06|
>
> **Table 4: Results of ablation experiments for w3.**
> |Dataset\w3|0|10|20|30|
> |:-:|:-:|:-:|:-:|:-:|
> |OBQA|26.00|31.40|31.00|35.40|
> |SIQA|44.32|47.95|48.93|51.54|
> |ARC-c|37.80|40.27|39.16|40.70|
> |WebQs|16.88|21.90|21.95|23.82|

---

### Note · Authors · 2025-12-29

I have read and agree with the venue's withdrawal policy on behalf of myself and my co-authors.